# Differentiable Trajectory Optimization as a Policy Class for Reinforcement and Imitation Learning

## Abstract

This paper introduces DiffTOP, a new policy class for reinforcement learning and imitation learning that utilizes differentiable trajectory optimization to generate the policy actions. Trajectory optimization is a powerful and widely used algorithm in control, parameterized by a cost and a dynamics function. The key to our approach is to leverage the recent progress in differentiable trajectory optimization, which enables computing the gradients of the loss with respect to the parameters of trajectory optimization. As a result, the cost and dynamics functions of trajectory optimization can be learned end-to-end, e.g., using the policy gradient loss in reinforcement learning, or using the imitation loss in imitation learning. When applied to model-based reinforcement learning, DiffTOP addresses the "objective mismatch" issue of prior algorithms, as the dynamics model in DiffTOP is learned to directly maximize task performance by differentiating the policy gradient loss through the trajectory optimization process. When applied to imitation learning, DiffTOP performs test-time trajectory optimization to compute the actions with a learned cost function, outperforming prior methods that only perform forward passes of the policy network to generate actions. We benchmark DiffTOP on 15 model-based RL tasks, and 13 imitation learning tasks with high-dimensional image and point cloud inputs, and show that it outperforms prior state-of-the-art methods in both domains.

## 1 Introduction

Recent works have shown that the representation of a policy can have a substantial impact on the learning performance (Chi et al., 2023; Florence et al., 2022; Amos et al., 2018; Seita et al., 2023). Prior works have explored the use of feed-forward neural networks (Seita et al., 2023), energy-based models (Florence et al., 2022), diffusion (Chi et al., 2023), or linear-quadratic regularizer (Amos et al., 2018) as the policy representation in the setting of imitation learning. In this paper, we propose DiffTOP, a new policy class which leverages **Diff**erentiable **T**rajectory **OP**timization to generate actions for reinforcement learning (RL) and imitation learning (IL).

Trajectory optimization is an effective and widely used algorithm in control, usually defined with a cost function and a dynamics function. In this paper, we view trajectory optimization as a policy class, where the parameters of the policy specify the cost function and the dynamics function, e.g., as neural networks. Given the learned cost and dynamics functions as well as the input state (e.g., images, point clouds, robot joint states), the policy then computes the actions by solving the trajectory optimization problem.

To apply such a policy to either RL or IL, we need to compute the gradients of the actions with respect to the policy parameters, which requires back-propagating through the trajectory optimization process. In this work, we leverage a recently developed software library, Theseus (Pineda et al., 2022), which is an efficient application-agnostic open source library for differentiable nonlinear least squares (DNLS) optimization built on PyTorch, to reliably differentiate through the trajectory optimization process. With Theseus, we are able to scale up DiffTOP to very high-dimensional states such as images and point clouds.

When applied to RL, DiffTOP computes the policy gradient loss on the generated actions from trajectory optimization. DiffTOP then differentiates through the trajectory optimization process to learn the dynamics and cost functions. This addresses the "objective mismatch" issue (Lambert et al., 2020; Eysenbach et al., 2022) of current model-based RL algorithms, i.e. models that achieve better training performance (e.g., lower MSE) in learning a dynamics model are not necessarily better for control. DiffTOP addresses this issue, as the latent dynamics and reward models are both optimized to maximize the task performance by back-propagating the policy gradient loss through the trajectory optimization process. We show that DiffTOP outperforms prior state-of-the-art model-based RL algorithms on 15 tasks from the DeepMind Control Suite (Tassa et al., 2018) with high-dimensional image inputs.

We also apply DiffTOP to imitation learning, which trains using a loss between the policy actions and the expert actions. Instead of outputting the policy actions directly, DiffTOP performs imitation learning by learning a cost function and performing test-time optimization with it. Using this approach, DiffTOP outperforms other types of policy classes that only perform forward passes of the policy network at test time. Relatedly, prior work (Florence et al., 2022) has explored learning an energy-based model for test-time optimization; however, we observe that our training procedure using differentiable trajectory optimization leads to better performance compared to the EBM approach used in prior work, which can suffer from training instability due to the requirement of sampling high-quality negative examples (Chi et al., 2023). We also outperform diffusion-based approaches (Chi et al., 2023) due to our procedure of learning a cost function that we optimize at test time. We show that DiffTOP achieves state-of-the-art performance for imitation learning across 13 different tasks on two widely used benchmarks, Robomimic (Mandlekar et al., 2021) (with image inputs) and Maniskill1 (Mu et al., 2021) and Maniskill2 (Gu et al., 2023) (with point cloud inputs).

In summary, the contributions of our paper are as following:

- We propose DiffTOP, a new policy class that uses differentiable trajectory optimization for reinforcement learning and imitation learning.
- We conduct extensive experiments to compare DiffTOP against prior state-of-the-art methods on 15 tasks for model-based RL and 13 tasks for imitation learning with high-dimensional sensory observations, and show that DiffTOP achieves state-of-the-arts results in both domains.
- We perform analysis and ablations of DiffTOP to provide insights into its learning procedure and performance gains.

## 2    RELATED WORKS

**Model-based reinforcement learning:** Compared to model-free RL, model-based RL usually has higher sample efficiency since it is solving a simpler supervised learning problem when learning the dynamics model. Recently, researchers have identified a fundamental problem for model-based RL, known as "objective mismatch" (Lambert et al., 2020). Some recent works have proposed a joint objective for model and policy learning in model-based RL, and the proposed objective is a lower bound on the true return of the policy (Eysenbach et al., 2022; Ghugare et al., 2022). In contrast to these works, we use Theseus (Pineda et al., 2022) to analytically compute the gradient of the true objective for updating the model.

From another view, we are treating the trajectory optimization procedure as an implicit policy. End-to-end MPC (Amos et al., 2018; Amos & Yarats, 2020) has been explored before as well, but they only test it in the imitation learning setting, and only on very low-dimensional control problems.

**Policy architecture for imitation learning:** Imitation learning can be formulated as the supervised regression task of learning to map observations to actions from demonstrations. Some recent work explores different policy architectures (e.g., explicit policy, implicit policy (Florence et al., 2022), diffusion policy (Chi et al., 2023)) and different action representations (e.g., mixtures of Gaussian (Bishop, 1994; Mandlekar et al., 2021), spatial action maps (Wu et al., 2020), action flow (Seita et al., 2023), or parameterized action spaces (Hausknecht & Stone, 2015)) to achieve more accurate learning from demonstrations, to model the multimodal distributions of demonstrations, and to capture sequential correlation. Our method distinguishes itself from the explicit or diffusion policy approaches in that we employ test-time optimization. In comparison with the implicit policy,

which also employs test-time optimization, we use a different and more stable training objective and procedure via differentiable trajectory optimization.

# 3 BACKGROUND

## 3.1 DIFFERENTIABLE TRAJECTORY OPTIMIZATION

In robotics and control, trajectory optimization solves the following type of problems:

$$\min_{a_0,\ldots,a_T} \sum_{t=0}^{T-1} c(s_t, a_t) + C(s_T) \tag{1}$$
$$s.t. \quad s_{t+1} = d(s_t, a_t)$$

where $c(s_t, a_t)$ and $C(s_T)$ are the cost functions, and $s_{t+1} = d(s_t, a_t)$ is the dynamics function. In this paper, we consider the case where the cost function and the dynamics functions are neural networks parameterized by $\theta$: $c_\theta(s_t, a_t)$, $C_\theta(s_T)$, and $d_\theta(s_t, a_t)$.

Let $a_0(\theta), \ldots, a_T(\theta)$ be the optimal solution to the trajectory optimization problem, which is a function of the model parameters $\theta$. Differentiable trajectory optimization is a class of method that enables fast and reliable computation of the gradient of the actions with respect to the model parameters $\frac{\partial a_t(\theta)}{\partial \theta}$. Specifically, in this paper we use Theseus (Pineda et al., 2022), which is an efficient application-agnostic open source library for differentiable nonlinear least squares optimization. Theseus works well with high-dimensional states, e.g., images or point clouds, along with using neural networks as the cost and dynamics functions.

## 3.2 MODEL-BASED RL PRELIMINARIES

We use the standard MDP formulation: $\langle \mathcal{S}, \mathcal{A}, \mathcal{R}, \mathcal{T}, \gamma \rangle$ where $\mathcal{S}$ is the state space, $\mathcal{A}$ is the action space, $\mathcal{R}(s, a)$ is the reward function, $\mathcal{T}(\cdot|s, a)$ is the transition dynamics function, and $\gamma \in [0, 1)$ is the is the discount factor. The goal is to learn a policy $\pi$ to maximize the expected return: $\mathbb{E}_{s_t, a_t \sim \pi}[\sum_{t=1}^{\infty} \gamma^t R(s_t, a_t)]$. In this paper we work on problems where the state space $S$ are high-dimensional sensory observations, e.g., images or point clouds. Model-based RL algorithms first learn a dynamics model, and then use it for learning a policy. When applied to model-based RL, our method builds upon TD-MPC (Hansen et al., 2022), a recently proposed model-based RL algorithm which we review briefly here. TD-MPC consists of the following components: first, an encoder $h_\theta$, which encodes the high-dimensional sensory observations, e.g., images, into a low-dimensional state $z_t = h_\theta(s_t)$. In the latent space, a latent dynamics model $d_\theta$ is also learned: $z_{t+1} = d_\theta(z_t, a_t)$. A latent reward predictor $R_\theta$ is learned which predicts the task reward $r$: $\hat{r} = R_\theta(z_t, a_t)$. Finally, a value predictor $Q_\theta$ learns to predict the Q value: $\hat{Q} = Q_\theta(z_t, a_t)$. Note that we use $\theta$ to denote all learnable parameters including the encoder, the latent dynamics model, the reward predictor, and the Q value predictor. These models are trained jointly using the following objective:

$$\mathcal{L}_{TD-MPC}(\theta; \tau) = \sum_{i=t}^{t+H} \lambda^{i-t} \mathcal{L}_{TD-MPC}(\theta; \tau_i), \tag{2}$$

where $\tau \sim \mathcal{B}$ is a trajectory $(s_t, a_t, r_t, s_{t+1})_{t:t+H}$ sampled from a replay buffer $\mathcal{B}$, $\lambda \in \mathbb{R}_+$ is a constant that weights near-term predictions higher, and the single-step loss is:

$$\mathcal{L}_{TD-MPC}(\theta; \tau_i) = c_1 \underbrace{\| R_\theta(\mathbf{z}_i, \mathbf{a}_i) - r_i \|_2^2}_{\text{reward}} + c_2 \underbrace{\| Q_\theta(\mathbf{z}_i, \mathbf{a}_i) - (r_i + \gamma Q_{\theta^-}(\mathbf{z}_{i+1}, \pi_\theta(\mathbf{z}_{i+1}))) \|_2^2}_{\text{value}}$$
$$+ c_3 \underbrace{\| d_\theta(\mathbf{z}_i, \mathbf{a}_i) - h_{\theta^-}(\mathbf{s}_{i+1}) \|_2^2}_{\text{latent state consistency}} \tag{3}$$

where $\theta^-$ are parameters of target networks that are periodically updated using the parameters of the learning networks. As shown in Equation 3, the parameters $\theta$ is optimized with a set of surrogate losses (reward prediction, value prediction, and latent consistency), rather than directly optimizing the task performance, known as the objective mismatch issue. At test time, model predictive path integral (MPPI) (Williams et al., 2016) is used for planning actions that maximize the predicted rewards and Q functions in the latent space. A policy $\pi_\psi$ is further learned in the latent space using the latent Q-value function, which is used to generate action samples in the MPPI process.

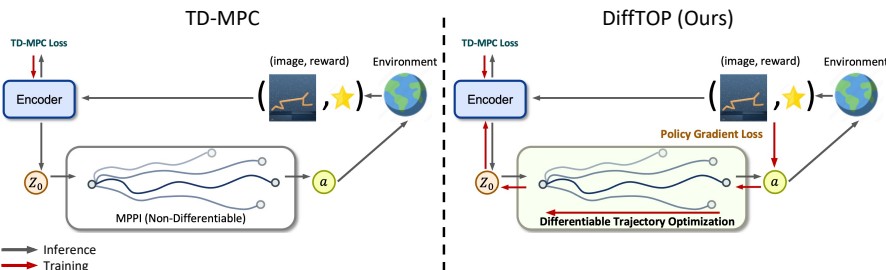

Figure 1: **Overview of DiffTOP for model-based RL**. In contrast to prior work in model-based RL Hansen et al. (2022) that uses non-differentiable MPPI (left), we utilize differentiable trajectory optimization to generate actions (right). DiffTOP computes the policy gradient loss on the generated actions and back-propagates it through the optimization process, which optimizes the encoder as well as the other latent space models (latent reward predictor and latent dynamics function) to maximize task performance.

## 4 METHOD

### 4.1 OVERVIEW

The core idea of our method DiffTOP is to use trajectory optimization as the policy $\pi_\theta$, where $\theta$ represents the parameters for the dynamics and cost functions. Given a state $s$, DiffTOP generates the actions $a(\theta)$ by solving the trajectory optimization problem in Equation 1 with $s_0 = s$. To optimize the policy parameters $\theta$, we use differentiable trajectory optimization to compute the gradients of the loss $\mathcal{L}(a(\theta))$ with respect to the policy parameters: $\frac{\partial \mathcal{L}(a(\theta))}{\partial \theta}$, where the exact form of the loss depends on the problem setting.

An overview of applying DiffTOP to model-based RL is shown in Figure 1. Existing model-based RL algorithms such as TD-MPC suffer from the objective mismatch issue: the latent dynamics and reward (cost) functions are learned to optimize a set of surrogate losses (as in Equation 3), instead of optimizing the task performance directly. DiffTOP addresses this issue: by computing the policy gradient loss on the optimized actions from trajectory optimization and differentiating through the trajectory optimization process, the dynamics and cost functions are optimized directly to maximize the task performance. We describe DiffTOP for model-based RL in Section 4.2.

We also apply DiffTOP to imitation learning; an overview is shown in Figure 2. In contrast to explicit policies that generate actions at test-time by forward passes of the policy network, DiffTOP generates the actions via test-time trajectory optimization with a learned cost function. This is in the same spirit of implicit behaviour cloning (Florence et al., 2022) which learns an energy function and optimizes with respect to it to generate actions at test-time. However, we observe that our training procedure using differentiable trajectory optimization leads to better performance compared to the EBM approach used in prior work, which can suffer from training instability due to the requirement of sampling high-quality negative examples (Chi et al., 2023). We describe DiffTOP for imitation learning in detail in Section 4.3.

### 4.2 DIFFERENTIABLE TRAJECTORY OPTIMIZATION APPLIED TO MODEL-BASED RL

We build DiffTOP on top of TD-MPC for model-based RL. Similar to TD-MPC, DiffTOP consists of an encoder $h_\theta$, a latent dynamics model $d_\theta$, a reward predictor $R_\theta$, and a Q-value predictor $Q_\theta$ (see Sec. 3.2). Note that we use $\theta$ to denote all learnable model parameters to be optimized in DiffTOP, including the parameters of the encoder $h_\theta$, the latent dynamics model $d_\theta$, the reward predictor $R_\theta$, and the Q value predictor $Q_\theta$. As shown in Figure 1, the key to DiffTOP is to change the non-differentiable MPPI planning algorithm in TD-MPC to a differentiable trajectory optimization, and include the policy gradient loss on the generated actions to optimize the model parameters $\theta$ directly for task performance.

Formally, given a state $s_t$, we use the encoder $h_\theta$ to encode it to the latent state $z_t$, and then construct the following trajectory optimization problem in the latent space:

$$a(\theta) = \underset{a_t, \ldots, a_{t+H}}{\arg\max} \sum_{l=t}^{H-1} \gamma^{l-t} R_\theta(z_t, a_t) + \gamma^H Q_\theta(z_H, a_H)$$
$$s.t. \ z_{t+1} = d_\theta(z_t, a_t)$$
(4)

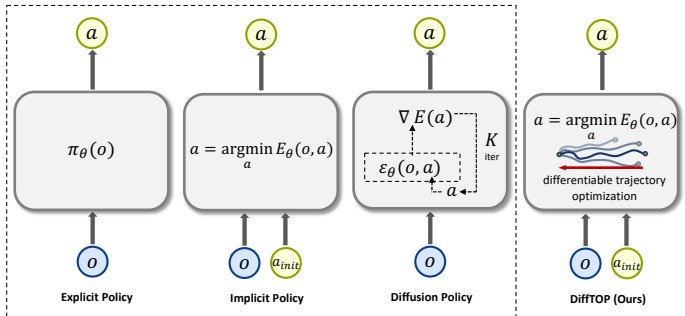

Figure 2: **Overview of our method on Imitation Learning.** DiffTOP (right) learns a cost function via differentiable trajectory optimization and performs test-time optimization with it, which is different from prior work (left) that uses an explicit policy or diffusion without test-time optimization. Although implicit policy shares the same spirit as DiffTOP, we observe that the training procedure of DiffTOP using differentiable trajectory optimization leads to better performance compared to the EBM approach used in prior work Florence et al. (2022), which can suffer from training instability.

where $H$ is the planning horizon. In this paper we leverage Theseus (Pineda et al., 2022) to solve Equation 4 in a differentiable way. Since Theseus only supports solving non-linear least-square optimization problems without constraints, we remove the dynamics constraints in the above optimization problem by manually rolling out the dynamics into the objective function. For example, with a planning horizon of $H = 2$, we turn the above optimization problem into the following one:

$$a(\theta) = \arg\max_{a_t, a_{t+1}, a_{t+2}} R_\theta(z_t, a_t) + R_\theta(d_\theta(z_t, a_t), a_{t+1}) + Q_\theta(d_\theta(d_\theta(z_t, a_t), a_{t+1}), a_{t+2}) \quad (5)$$

We set the values of $H$ following the schedule as in TD-MPC, and we use the Levenberg–Marquardt algorithm in Theseus to solve the optimization problem. Following TD-MPC, we also learn a policy $\pi_\psi$ in the latent space using the learned Q-value predictor $Q_\theta$, and the output from the policy is used as the action initialization for solving Equation 4.

Let $a(\theta)$ be the solution of the above trajectory optimization problem, obtained using Theseus as described above. DiffTOP is learned with the following objective, which jointly optimizes the encoder, latent dynamics model, latent reward model, and the Q-value predictor:

$$\mathcal{L}_{DiffTOP}^{RL}(\theta; \tau) = \sum_{i=t}^{t+H} \lambda^{i-t} \left( \mathcal{L}_{TD-MPC}(\theta; \tau_i) + c_0 \mathcal{L}_{PG}(\theta; \tau_i) \right) \quad (6)$$

$$\mathcal{L}_{PG}(\theta; \tau_i) = \tilde{Q}_\phi(s_i, a(\theta))$$

where $\tilde{Q}_\phi$ is the Q function learned via Bellman updates (Watkins & Dayan, 1992) which is used to compute the deteministic policy gradient (Lillicrap et al., 2015), and $c_0$ is the weight for this loss term. $\tilde{Q}_\phi$ is learned in the original state space $\mathcal{S}$ instead of the latent space to provide accurate policy gradients. The key idea here is that we can backpropagate through the policy gradient loss $\mathcal{L}_{PG}$, which backpropagates through $a(\theta)$ and then through the differentiable trajectory optimization procedure of Equation 4 to update $\theta$.

### 4.3 DIFFERENTIABLE TRAJECTORY OPTIMIZATION APPLIED TO IMITATION LEARNING

We also use DiffTOP for model-based imitation learning. A comparison of DiffTOP to other types of policy classes used in prior work is shown in Figure 2. In this approach, DiffTOP consists of an encoder $h_\theta$ and a latent dynamics function $d_\theta$, as before. However, in the setting of imitation learning, we do not assume access to a reward function $\mathcal{R}(s, a)$. Instead, we generate actions by solving the following trajectory optimization problem:

$$a(\theta) = \arg\max_{a_t, \dots, a_{t+H}} \sum_{l=t}^{H} \gamma^{l-t} f_\theta(z_t, a_t) \quad (7)$$

$$s.t. \ z_{t+1} = d_\theta(z_t, a_t),$$

in which $f_\theta(z_t, a_t)$ is a function over the latent state $z_t$ and actions $a_t$ that we will optimize using the imitation learning loss, as described below. Similarly, We use $\theta$ to denote all learnable model

parameters to be optimized in DiffTOP, which includes the parameters of the encoder $h_\theta$, the latent dynamics model $d_\theta$, and the function $f_\theta$ in the imitation learning setting.

In imitation learning, we assume access to an expert dataset $D = \{(s_i, a_i^*)\}_{i=1}^N$ of state-action pairs $(s_i, a_i^*)$. In the most basic form, the loss $\mathcal{L}$ for DiffTOP can be the mean square error between the the expert actions $a_i^*$ and the actions $a(\theta)$ returned from solving Equation 7:

$$\mathcal{L}_{BC}(\theta) = \sum_{i=1}^N ||a(\theta) - a_i^*|| \tag{8}$$

The key idea here is that we can backpropagate through the imitation loss $\mathcal{L}_{BC}$, which backprop-agates through $a(\theta)$ and then through the differentiable trajectory optimization procedure of Equation 7 to update $\theta$. This enables us to learn the function $f_\theta(z_t, a_t)$ used in the optimization Equation 7 directly by optimizing the imitation loss $\mathcal{L}_{BC}(\theta)$. Because this loss is optimized through the trajectory optimization procedure (Equation 7), we will learn a function $f_\theta(z_t, a_t)$ such that optimizing Equation 7 returns actions that match the expert actions.

**Multimodal DiffTOP:** The loss in Equation 8 will not be able to capture multi-modal action distributions in the expert demonstrations. To address this, we use a Conditional Variational Auto-Encoder (CVAE) (Sohn et al., 2015) as the policy architecture, which has the ability to capture a multi-modal action distribution (Zhao et al., 2023). The CVAE encodes the state $s_i$ and the expert action $a_i^*$ into a latent vector $z_i$; the decoder takes as input a sampled latent $z_i$ and the state $s_i$ to decode the action $a(\theta)$.

The key idea in our our approach is that the decoder takes the form of a trajectory optimization algorithm, given by Equation 7. This algorithm takes as input the latent $z_i$ and the state $s_i$ and uses differentiable trajectory optimization (e.g., Theseus) to decode the action $a(\theta)$. Because this trajectory optimization is differentiable, we can backpropagate through it to learn the parameters $\theta$ for the encoder, dynamics $d_\theta$, and the function $f_\theta$ used in Equation 7. See Appendix D for further details.

**Action refinement:** We also note that DiffTOP provides a natural way to perform action refinement on top of a base policy. Given an action from any base policy, we can use this action as the initialization of the action variables for solving the trajectory optimization problem; the trajectory optimizer will iteratively refine this action initialization with respect to the optimization objective of Equation 7. In our experiments, we find DiffTOP always outperforms the base policies when using their actions as the initialization, and it also outperforms other ways of performing action refinement, such as residual learning.

## 5 EXPERIMENTS

### 5.1 MODEL-BASED REINFORCEMENT LEARNING

We conduct experiments on 15 DeepMind Control suite tasks, which involve simulated locomotion and manipulation tasks, such as making a cheetah run or swinging a ball into a cup. All tasks use image observations and the control policy does not have direct access to the underlying states.

We compare to the following baselines: **TD-MPC** (Hansen et al., 2022), a state-of-the-art model-based RL algorithm, which DiffTOP builds on. **Dreamer-v2** (Hafner et al., 2020), another state-of-the-art model-based RL algorithm that has an image reconstruction loss when learning the latent state space. **Dreamer-v3** (Hafner et al., 2023), an upgraded version of Dreamer-v2 with better results on many tasks. **DrQ-v2** (Yarats et al., 2021), a state-of-the-art model-free RL algorithm.

Figure 3 shows the learning curves for all methods on all tasks. The top-left subplot shows the nor-malized performance averaged across all 15 tasks, which is computed as the achieved return divided by the max return from any algorithm. As shown, DiffTOP (red curve) outperforms all compared baselines, and establishes a new state-of-the-art performance for RL on DeepMind Control Suite. We especially note that the performance of DiffTOP is much higher than TD-MPC, which DiffTOP builds on, showing the benefit of adding the policy gradient loss and directly differentiating through it to optimize the learned latent spaces. Compared to Dreamer-v3, the state-of-the-art model-based RL algorithm that has been heavily tuned, DiffTOP learns faster in early stages and achieves similar final performance. We also note that Dreamer-v3 uses a more complicated network architecture (i.e.,

the recurrent state space model (RSSM (Hafner et al., 2019)) than DiffTOP, which uses a simpler latent space model inherited from TD-MPC. We leave incorporating DiffTOP with more advanced latent space models as future work, which we believe might further boost the performance. We present results on computational efficiency (return vs wall-clock time) of DiffTOP in Appendix A.1.

We also perform ablation studies to examine how each loss term in Equation 6 contributes to the final performance of DiffTOP. The results are shown in Figure 4. We find that removing the reward prediction loss causes DiffTOP to completely fail. Removing the dynamics loss, or not using the action initialization from the learned policy $\pi_\psi$ for solving the trajectory optimization, both lead to a decrease in the performance. These shows the necessity of using all the loss terms in DiffTOP for learning a good latent space to achieve strong performance.

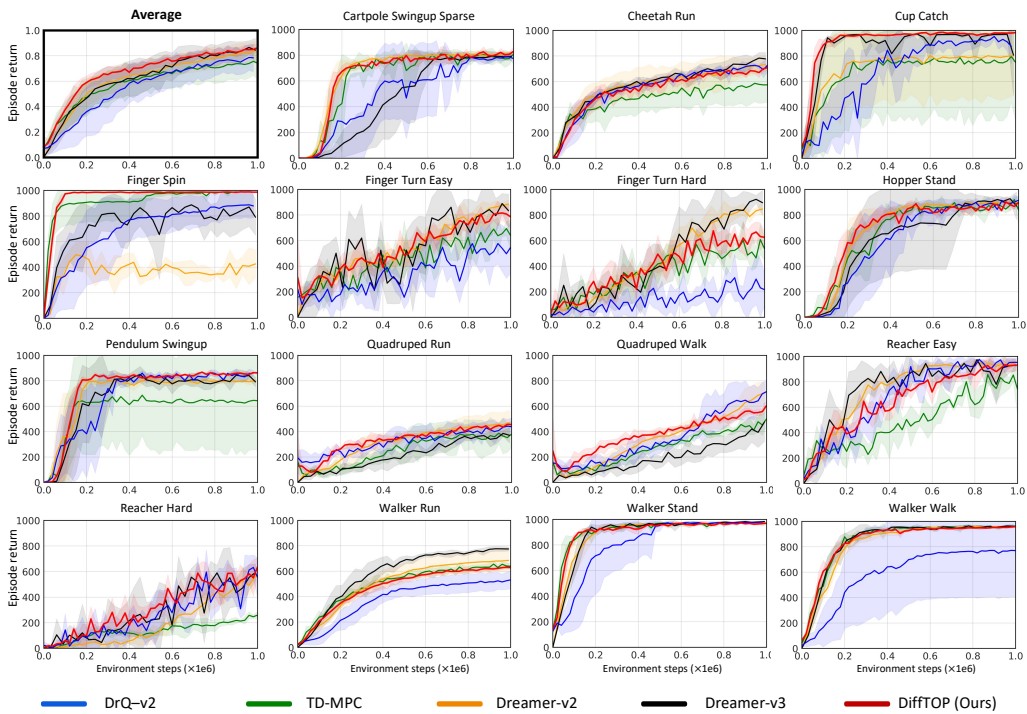

Figure 3: Performance of DiffTOP, in comparison to 4 prior state-of-the-art model-based and model-free RL algorithms, on 15 tasks from DeepMind control suite. DiffTOP achieves the best performance when averaged across all tasks, and learns faster in early stages compared to Dreamer-v3. Results are averaged with 4 seeds, and the shaded regions represent the standard deviation.

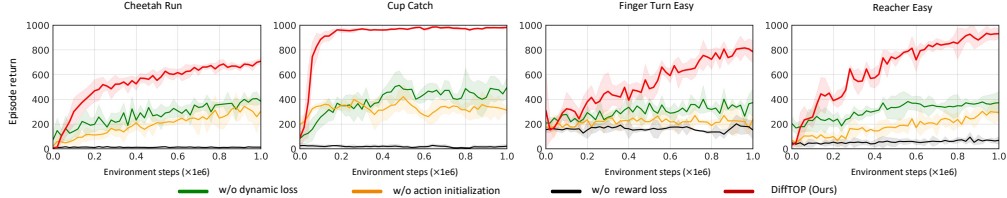

Figure 4: Ablation study of DiffTOP to examine the contribution of each loss terms towards the final performance, on a subset of 4 tasks. We find the reward prediction loss, action initialization, and dynamics prediction loss are all essential for DiffTOP to achieve good performance.

## 5.2 IMITATION LEARNING

### 5.2.1 ROBOMIMIC

Robomimic (Mandlekar et al., 2021) is a large-scale benchmark designed to study imitation learning for robot manipulation. The benchmark encompasses a total of 5 tasks with two types of demonstrations: collected from proficient humans (PH) or a mixture of proficient and non-proficient humans.

We use the PH demonstrations, and evaluate on three of the most challenging tasks: Square, Transport, and ToolHang. We use image-based observations and the default velocity controller for all the tasks. In addition to Robomimic, we compare to another task, Push-T from the diffusion policy (Chi et al., 2023) task set, to demonstrate that we can learn multimodal cost functions by using the CVAE training loss.

| | IBC | BC-RNN | Residual +BC-RNN | DiffTOP (Ours) + BC-RNN | Diffusion | IBC + Diffusion | Residual + Diffusion | DiffTOP (Ours) + Diffusion |
|---|---|---|---|---|---|---|---|---|
| Square | 0.04±0.00 | 0.82±0.00 | 0.84±0.01 | 0.90±0.02 | 0.88±0.03 | 0.68±0.05 | 0.88±0.02 | **0.92**±0.01 |
| Transport | 0.00±0.00 | 0.72±0.03 | 0.74±0.03 | 0.83±0.02 | 0.93±0.04 | 0.08±0.03 | 0.92±0.01 | **0.96**±0.01 |
| ToolHang | 0.00±0.00 | 0.67±0.04 | 0.72±0.03 | 0.82±0.00 | 0.90±0.00 | 0.06±0.01 | 0.90±0.00 | **0.92**±0.01 |
| Push-T | 0.11±0.01 | 0.70±0.02 | 0.72±0.02 | 0.75±0.02 | **0.91**±0.00 | 0.08±0.01 | **0.91**±0.00 | **0.91**±0.01 |

Table 1: Comparison of DiffTOP with all other mehtods on the Robomimic tasks. DiffTOP achieves the best performances on all tasks when using diffusion policy as the base policy.

We compare to the following baselines: **IBC** (Florence et al., 2022): An implicit policy that learns an energy function conditioned on both action and observation using the InfoNCE loss (Oord et al., 2018). **BC-RNN** (Mandlekar et al., 2021): A variant of BC that uses a Recurrent Neural Network (RNN) as the policy network to encode a history of observations. This is the best-performing baseline in the original Robomimic (Mandlekar et al., 2021) paper. **Residual + BC-RNN**: We use a pretrained BC-RNN as the base policy, and learn a residual policy on top of it. The residual policy takes as input the action from the base policy, and outputs a delta action which is added to the base action. This is the most standard and simple way of doing residual learning. **Diffusion Policy** (Chi et al., 2023): A policy that uses the diffusion model as the policy class. It refines noise into actions via a learned gradient field. **IBC + Diffusion**: A version of IBC that uses the action from a pretrained Diffusion Policy as the action initialization in the test-time optimization process. **Residual + Diffusion**: Similar to Residual + BC-RNN, but using a pre-trained Diffusion Policy as the base policy. For DiffTOP, we compare two variants of it: DiffTOP + BC-RNN and DiffTOP + Diffusion Policy, which uses a pre-trained BC-RNN or a pre-trained diffusion policy as the base policy to generate the initialization action for solving the trajectory optimization problem. In Appendix A.2, we also present results of DiffTOP with zero initialization or random initialization, instead of initializing the action from a base policy.

The results are shown in Table 1. We find that DiffTOP+Diffusion Policy achieves the highest success rates consistently across all tasks. Furthermore, irrespective of the base policy used — whether BC-RNN or Diffusion Policy — DiffTOP always brings noticeable improvement in the performance over the base policy. While learning a residual policy does lead to improvements upon the base policy, DiffTOP shows a significantly greater performance boost. In addition, by comparing DiffTOP+Diffusion Policy with IBC+Diffusion Policy, we find that using the same action initialization for IBC is considerably less effective than using the same action initialization in DiffTOP. In many tasks, even when the base Diffusion Policy already exhibits high success rates, IBC+Diffusion Policy still results in poor performances, indicating the training objective used in IBC actually deteriorates the base actions.

Note that for the three tasks in Table 1 from Robomimic, we use the default velocity controller from Robomimic. We note the use of the velocity controller leads to a small decline in the performance of the Diffusion Policy compared to its performance in the original paper where a positional controller is used. Results for using the positional controller can be found in the appendix, where our method performs on par or slightly better than diffusion policy, since the performance of diffusion policy has almost saturated with a positional controller. The Push-T task still uses the default position controller as in the diffusion policy paper.

We also show the benefit of using a CVAE architecture for DiffTOP, which enables DiffTOP to capture multimodal action distributions. In our case, with different latent samples from CVAE, we get different objective functions $f_\theta(z, a)$ and dynamics functions $d_\theta(z, a)$, allowing DiffTOP to generate different actions from the same state. Figure 5 illustrates the multimodal objective function learned by DiffTOP (right), and the resulting multimodal actions (left). The left subplot shows that when starting from the same action initialization $a_{init}$, with two different latent samples, DiffTOP optimizes $a_{init}$ into two different actions, $\hat{a}_1$ and $\hat{a}_2$ that move in distinct directions. The trajectory optimization procedure that iteratively updates the action is represented by dashed lines transitioning from faint to solid. From these two actions, two distinct trajectories are subsequently generated to push the T-shape object towards its goal. The middle and right subplots show the objective function

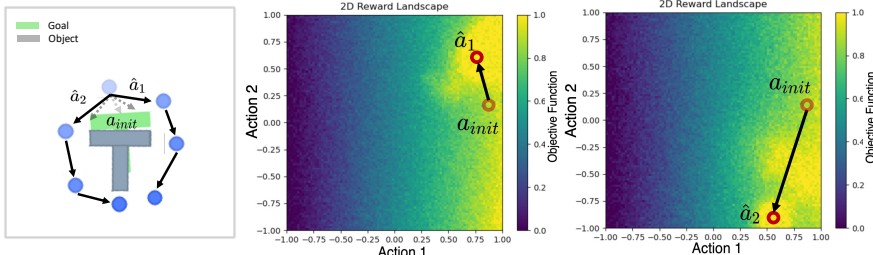

Figure 5: By using a CVAE, DiffTOP can learn multimodal objectives functions via sampling different latent vectors from CVAE (right). By performing trajectory optimization with these two different objective functions, DiffTOP can generate multimodal actions (left).

landscapes for the 2 different samples, as well as the initial action $a_{init}$, and the final optimized action $\hat{a_1}$ and $\hat{a_2}$. We note the two landscapes are distinct from each other with different optimal solutions, demonstrating that DiffTOP can generate multimodal objective functions and thus capture multimodal action distributions. We note that the learned objective function $f$ is not necessarily a "reward" function as those learned via inverse RL Ng et al. (2000). It is just a learned "objective function", such that optimizing it with trajectory optimization would yield actions that minimize the imitation learning loss with respect to the expert actions in the demonstration. We leave exploring the connections with inverse RL for future work.

### 5.2.2 MANISKILL

ManiSkill (Mu et al., 2021; Gu et al., 2023) is a unified benchmark for learning generalizable robotic manipulation skills with 2D & 3D visual input. It includes a series of rigid body tasks (e.g., Pick-Cube, PushChair) and soft body tasks (e.g., Fill, Pour). We choose 9 tasks (4 soft body tasks and 5 rigid body tasks) from ManiSkill1 (Mu et al., 2021) and ManiSkill2 (Gu et al., 2023) and use 3D point cloud input for all the tasks. We use the end-effector frame as the observation frame (Liu et al., 2022) and use the PD controller with the end-effector delta pose as the action.

We build our method on top of the strongest imitation learning baseline in ManiSkill2, which is a Behavior Cloning (BC) policy with PointNet (Qi et al., 2017) as the encoder. Again, we also compare to BC+residual, which learns a residual policy that takes as input the action from the BC policy and outputs a delta correction. The results are shown in Table 2. As shown, DiffTOP + BC consistently outperforms both baselines on all tasks, demonstrating the strong effectiveness of using differentiable trajectory optimization as the policy class.

| | PickCube | Fill | Hang | Excavate | Pour | OpenCabinet Drawer | OpenCabinet Door | PushChair | MoveBucket |
|---|---|---|---|---|---|---|---|---|---|
| BC | 0.19±0.03 | 0.72±0.04 | 0.76±0.02 | 0.25±0.02 | 0.13±0.01 | 0.47±0.03 | 0.35±0.04 | 0.12±0.01 | 0.10±0.01 |
| BC + residual | 0.21±0.04 | 0.75±0.02 | 0.75±0.02 | 0.27±0.03 | 0.12±0.01 | 0.49±0.02 | 0.36±0.03 | 0.15±0.02 | 0.10±0.01 |
| DiffTOP(Ours) + BC | **0.32**±0.02 | **0.82**±0.01 | **0.85**±0.03 | **0.29**±0.01 | **0.17**±0.02 | **0.53**±0.02 | **0.45**±0.02 | **0.20**±0.02 | **0.15**±0.02 |

Table 2: Comparison of all the methods on the Maniskill2 baseline. DiffTOP consistently outperforms both baselines on all tasks.

## 6 CONCLUSION AND DISCUSSION

We introduce DiffTOP, a new policy class for reinforcement learning and imitation learning that uses differentiable trajectory optimization to generate the policy actions. The key to our approach is to utilize the recent progress in differentiable trajectory optimization to enable computing the gradients of the loss with respect to the parameters of trajectory optimization, and learn the cost and dynamics functions of trajectory optimization end-to-end. When applied to model-based reinforcement learning, DiffTOP addresses the "objective mismatch" issue of prior methods, since the dynamics model in DiffTOP is learned to directly maximize task performance by differentiating the policy gradient loss through the trajectory optimization process. When applied to imitation learning, DiffTOP performs test-time trajectory optimization to compute the actions with a learned objective function, achieving better performances than prior methods that only perform forward passes of the policy network to generate actions. We benchmark DiffTOP on 15 model-based RL tasks, and 13 imitation learning tasks with image and point cloud inputs, and show that it greatly outperforms prior state-of-the-art methods in both domains.

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

# Appendix

## A   ADDITIONAL RESULTS

### A.1   MODEL-BASED REINFORCEMENT LEARNING

In model-based reinforcement learning, the key distinctions between DiffTOP and TD-MPC (Hansen et al., 2022) are: 1) TD-MPC employs the Model Predictive Path Integral (MPPI (Williams et al., 2015)) in the planning stage, whereas we utilize trajectory optimization. 2) In addition to the original loss used in TD-MPC, we use an additional policy gradient loss and back-propagate it through the differentiable trajectory optimization process to update the model parameters. Figure 6 shows that the improvement of DiffTOP over TD-MPC comes from the addition of the policy gradient loss, instead of purely changing MPPI to trajectory optimization. To be more specific, we compare TD-MPC with DiffTOP (w/o backward), a variant of DiffTOP that removes the policy gradient loss for updating the model parameters. The results indicate that TD-MPC and the DiffTOP (w/o backward) variant perform comparably, suggesting that using MPPI or trajectory optimization at test-time for action generation have similar performances. With the inclusion of the policy gradient loss, DiffTOPsignificantly outperforms both TD-MPC and the DiffTOP (w/o backward) variant, demonstrating the efficacy of adding the policy gradient loss in DiffTOP.

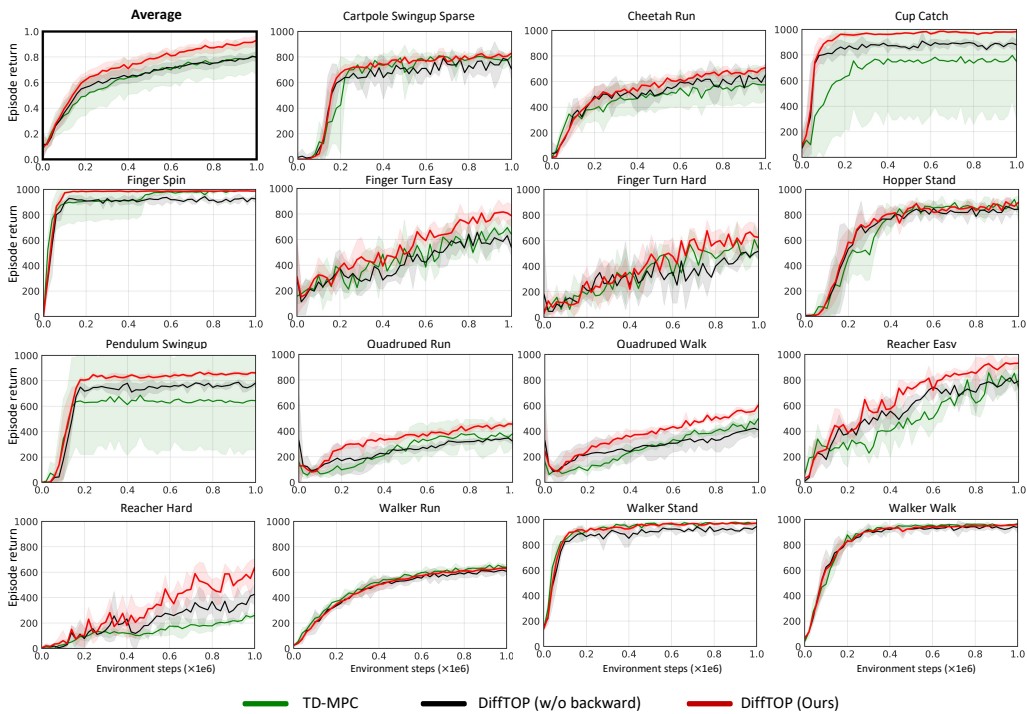

Figure 6: Performance of DiffTOP, in comparison to TD-MPC and DiffTOP (w/o backward) on 15 tasks from DeepMind control suite.

In addition to comparing the sample efficiency of DiffTOP to prior methods, we also compare the computational efficiency of DiffTOP versus TD-MPC on some of the environments. This is shown in Figure 7, where the y-axis is the return, and the x-axis is the wall-clock time used to train DiffTOP and TD-MPC for 1M environment steps. As shown, it takes more wall-clock time for DiffTOP to finish the training. In terms of computational efficiency, the results are environment-dependent. DiffTOP achieves better computational efficiency on reacher-hard and cup-catch. On pendum-swingup, TD-MPC converges to a sub-optimal value in the early training stage and DiffTOP outperforms it within 24 hours of training time. DiffTOP has similar computational efficiency on cartpole-swingup-sparse, reacher-easy, and finger-spin, and slightly worse computational efficiency

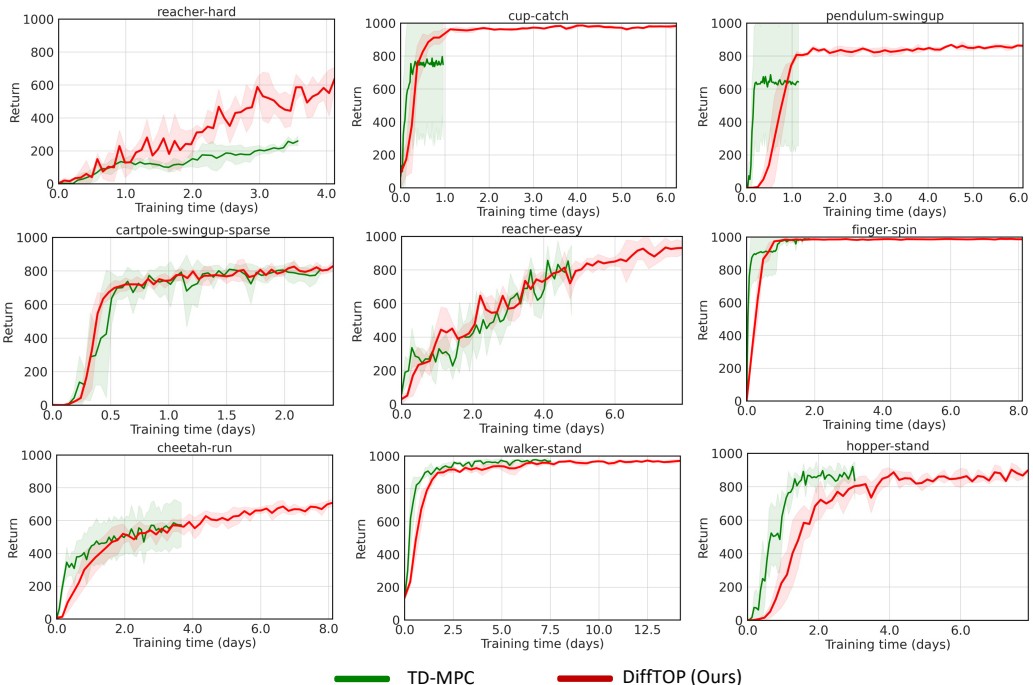

Figure 7: Return vs wall-clock time of DiffTOP and TD-MPC on some of the RL environments. The x-axis is the training time in days (24 hours), and the y-axis is the return. Both methods are trained for 1M environments steps. The training takes a long time (a few days on some environments) because the policy observation is high-dimensional images.

on cheetah-run and walker-stand compared to TD-MPC. The gap is larger on hopper-stand. The major reason for DiffTOP to take longer time for training is that solving and back-propagating through the trajectory optimization problem in Equation 4 is slower than doing MPPI as used in TD-MPC. As a reference, to infer the action at one time step, it takes $0.052$ second to use Theseus to solve and differentiate through the trajectory optimization problem in Equation 4, and $0.0092$ second for using MPPI in TD-MPC. However, we also want to note that the community is actively developing better and faster algorithms/software libraries for differentiable trajectory optimization, which could improve the computation efficiency of DiffTOP. For example, in all our experiments, we used the default CPU-based solver in Theseus. Theseus also provides a more advanced solver named BaSpaCho, which is a batched sparse Cholesky solver with GPU support. When we switch from the default CPU-based solver to BaSpaCho, the time cost of solving the trajectory optimization problem in Equation 4 is reduced by 22% from $0.052$ second to $0.041$ second. With better libraries/algorithms in the future for differentiable trajectory optimization, we believe the computational efficiency of DiffTOP would further improve.

## A.2 IMITATION LEARNING

We also present results of DiffTOP with zero initialization or random initialization, where instead of initializing the action from a base policy, the action is initialized to be 0, or randomly sampled from $\mathcal{N}(0, 1)$, on RoboMimic and Maniskill.

The results on RoboMimic is shown in Table 3. We notice a drop in performance of DiffTOP with zero or randomly-initialized actions, possibly due to the convergence to bad local minima during nonlinear trajectory optimization without a good action initialization. We note this would not be a drawback of applying DiffTOP in practice for imitation learning: one could always first learn a base policy using any behavior cloning algorithm, and then use DiffTOP to further refine the actions.

The results on Maniskill is shown in Table 4. Again, if we use zero or random action initialization with DiffTOP, the performance drops to be similar to or slightly worse than vanilla BC. Therefore,

we think a good practice of using DiffTOP for imitation learning would be to always try to provide it with a good action initialization, e.g., by first training a BC policy and use its action as the initialization in DiffTOP.

| | IBC | BC-RNN | Residual +BC-RNN | DiffTOP (Ours) + BC-RNN | Diffusion | IBC + Diffusion | Residual + Diffusion | DiffTOP (Ours) + Diffusion | DiffTOP (Ours) + zero init. | DiffTOP (Ours) + random init. |
|---|---|---|---|---|---|---|---|---|---|---|
| Square | 0.04±0.00 | 0.82±0.00 | 0.84±0.01 | 0.90±0.02 | 0.88±0.03 | 0.68±0.05 | 0.88±0.02 | **0.92**±0.01 | 0.84±0.02 | 0.80±0.00 |
| Transport | 0.00±0.00 | 0.72±0.03 | 0.74±0.03 | 0.83±0.02 | 0.93±0.04 | 0.08±0.03 | 0.92±0.01 | **0.96**±0.01 | 0.42±0.01 | 0.36±0.04 |
| ToolHang | 0.00±0.00 | 0.67±0.04 | 0.72±0.03 | 0.82±0.00 | 0.90±0.00 | 0.06±0.01 | 0.90±0.00 | **0.92**±0.01 | 0.00±0.00 | 0.00±0.00 |
| Push-T | 0.11±0.01 | 0.70±0.02 | 0.72±0.02 | 0.75±0.02 | **0.91**±0.00 | 0.08±0.01 | **0.91**±0.00 | **0.91**±0.01 | 0.62±0.04 | 0.57±0.02 |

Table 3: Comparison of DiffTOP with all other mehtods on the Robomimic tasks. DiffTOP achieves the best performances on all tasks when using diffusion policy as the base policy. If zero or random initialization are used in DiffTOP, the performance drops, possibly due to the convergence to bad local minima during nonlinear trajectory optimization without a good action initialization.

| | PickCube | Fill | Hang | Excavate | Pour | OpenCabinet Drawer | OpenCabinet Door | PushChair | MoveBucket |
|---|---|---|---|---|---|---|---|---|---|
| BC | 0.19±0.03 | 0.72±0.04 | 0.76±0.02 | 0.25±0.02 | 0.13±0.01 | 0.47±0.03 | 0.35±0.04 | 0.12±0.01 | 0.10±0.01 |
| BC + residual | 0.21±0.04 | 0.75±0.02 | 0.75±0.02 | 0.27±0.03 | 0.12±0.01 | 0.49±0.02 | 0.36±0.03 | 0.15±0.02 | 0.10±0.01 |
| DiffTOP(Ours) + BC | **0.32**±0.02 | **0.82**±0.01 | **0.85**±0.03 | **0.29**±0.01 | **0.17**±0.02 | **0.53**±0.02 | **0.45**±0.02 | **0.20**±0.02 | **0.15**±0.02 |
| DiffTOP (Ours) + zero init. | 0.20±0.03 | 0.76±0.03 | 0.72±0.02 | 0.25±0.01 | 0.04±0.00 | 0.50±0.04 | 0.34±0.04 | 0.04±0.01 | 0.06±0.00 |
| DiffTOP (Ours) + random init. ˙ | 0.18±0.02 | 0.68±0.03 | 0.67±0.01 | 0.19±0.04 | 0.04±0.00 | 0.39±0.04 | 0.30±0.02 | 0.00±0.00 | 0.05±0.01 |

Table 4: Comparison of all the methods on the Maniskill2 baseline. DiffTOP consistently outperforms both baselines on all tasks with action initialization from the BC policy. If zero or random initialization are used in DiffTOP, the performance drops, possibly due to the convergence to bad local minima during nonlinear trajectory optimization without a good action initialization.

In the original Diffusion Policy (Chi et al., 2023) paper, it was observed that the use of positional controllers yielded superior results for Diffusion Policy compared to the default velocity controller on Robomimic (Mandlekar et al., 2021) tasks. We evaluate Diffusion Policy, which is the strongest baseline, and DiffTOP on the most difficult three tasks with ph (proficient-human demonstration) and mh (multi-human demonstration) demonstrations using positional controller. The results with the positional controller are presented in Table 5. Diffusion Policy already achieves nearly the maximal possible performance on most tasks with the positional controller. DiffTOP, however, is able to achieve similar or even higher performances on most of these tasks.

| | Square (ph) | Square (mh) | Transport (ph) | Transport (mh) | ToolHang (ph) |
|---|---|---|---|---|---|
| Diffusion | **0.98**±0.01 | **0.97**±0.02 | **1.00**±0.00 | 0.88±0.02 | 0.95±0.02 |
| DiffTOP + Diffusion | **0.98**±0.01 | 0.96±0.02 | **1.00**±0.00 | **0.91**±0.01 | **0.96**±0.01 |

Table 5: Performance Comparison of DiffTOP and Diffusion Policy using Positional Controllers on Robomimic Tasks.

Additionally, we do ablation experiments on the planning horizon $H$ for imitation learning, with the results presented in Table 6. We observe that simply increasing the planning horizon $H$ in imitation learning does not necessarily enhance performance. As the horizon increases from $H = 1$ to $H = 3$, the performance remains nearly the same; however, when $H$ is increase to $5$, we observe a slight decline in the performance.

## B   IMPLEMENTATION DETAILS

In this section, we describe the implementation details of DiffTOP for the model-based RL experiments. For the imitation learning part, the code structure is very similar to this model-based RL implementation. For more detailed information, please refer to the code we will release upon acceptance of the paper. We implement DiffTOP on top of the open-source implementation of TD-MPC (Hansen et al., 2022) from the authors. Below we show the pseudo-code of the training function in DiffTOP.

| | Square (ph) | Transport (ph) | ToolHang (ph) | Push-T |
|---|---|---|---|---|
| $H = 1$ | **0.92**±0.01 | **0.96**±0.01 | **0.92**±0.01 | **0.91**±0.01 |
| $H = 3$ | **0.92**±0.01 | 0.94±0.02 | **0.92**±0.00 | 0.88±0.02 |
| $H = 5$ | 0.91±0.01 | 0.94±0.01 | 0.90±0.00 | 0.88±0.01 |

Table 6: Ablation experiments for the planning horizon $H$ in imitation learning.

```python
def train():
    """
    Training code
    """
    for step in range(total_steps):
        obs = env.reset()
        # Differentiable trajectory optimization and update model
        action, info = agent.plan_theseus_update(obs)
        # Env step
        obs, reward, done, _ = env.step(action.cpu().numpy())
        # collect data in buffer and update model (TD-MPC loss)
        replay_buffer += (obs, action, reward, done)
        agent.update(replay_buffer)
```

Then, we demonstrate how the policy gradient loss is computed by differentiable trajectory optimization in DiffTOP with PyTorch-like pseudocode:

```python
def plan_theseus_update(obs):
    """
    Differentiable trajectory optimization and update model using policy
    gradient loss.
    h, R, Q, d: model components.
    c0: loss coefficients.
    """
    import theseus as th

    # Encode first observation
    z = self.model.h(obs)

    # Get initialization action from pi
    init_actions = self.model.pi(z)

    # Theseus variable
    actions = th.Vector(tensor=actions, name="actions")
    obs = th.Variable(obs, name="obs")

    # Cost Function and Objective
    cost_function = th.AutoDiffCostFunction([obs], [action]
        ,value_cost_fn)
    objective = th.Objective().add(cost_function)

    # Trajectory optimization optimizer
    theseus_optim = th.TheseusLayer(th_optimizer)

    # Theseus layer forward
    theseus_inputs = {"actions": init_actions, "obs": obs}
    updated_inputs, info = theseus_optim.forward(theseus_inputs)
    updated_actions = updated_inputs['actions']

    # Update model using policy gradient losss
    a_loss = - torch.min(*self.model.Q_s(obs, updated_actions[0]))*c0
    a_loss.backward()
    optim_a.step()
```

For model-based reinforcement learning, We provide the network details for the added networks we used upon TD-MPC, which are the twin Q networks $\tilde{Q}_\phi$ learned in the original state space for computing the deterministic policy gradient.

```
(Q_s1): Sequential(
    (0): Linear(in_features=S, out_features=256)
    (1): ELU(alpha=1.0)
    (2): Linear(in_features=256, out_features=Z))
    (3): Linear(in_features=Z+A, out_features=512)
    (4): LayerNorm((512,), elementwise_affine=True)
    (5): Tanh()
    (6): Linear(in_features=512, out_features=512)
    (7): ELU(alpha=1.0)
    (8): Linear(in_features=512, out_features=1))
(Q_s2): Sequential(
    (0): Linear(in_features=S, out_features=256)
    (1): ELU(alpha=1.0)
    (2): Linear(in_features=256, out_features=Z))
    (3): Linear(in_features=Z+A, out_features=512)
    (4): LayerNorm((512,), elementwise_affine=True)
    (5): Tanh()
    (6): Linear(in_features=512, out_features=512)
    (7): ELU(alpha=1.0)
    (8): Linear(in_features=512, out_features=1))
```

For Imitation Learning, The default network details are as follows. Note that for Robomimic (Mandlekar et al., 2021) and Push-T tasks, we use the RNN-encoder from Robomimic; for ManiSkill (Mu et al., 2021; Gu et al., 2023) tasks, we use the PointNet encoder from ManiSkill2 Gu et al. (2023).

```
(ho): Sequential(
    (0): Linear(in_features=S, out_features=256)
    (1): ELU(alpha=1.0)
    (2): Linear(in_features=256, out_features=256)
    (3): ELU(alpha=1.0)
    (4): Linear(in_features=256, out_features=Zs))
(ha): Identity
(hl): Sequential(
    (0): Linear(in_features=Zs+A, out_features=256)
    (1): ELU(alpha=1.0)
    (2): Linear(in_features=256, out_features=256)
    (3): ELU(alpha=1.0)
    (4): Linear(in_features=256, out_features=128))
(R): Sequential(
    (0): Linear(in_features=Zs+A+64, out_features=512)
    (1): ELU(alpha=1.0)
    (2): Linear(in_features=512, out_features=512)
    (3): ELU(alpha=1.0)
    (4): Linear(in_features=512, out_features=1))
(d): Sequential(
    (0): Linear(in_features=Zs+A+64, out_features=512)
    (1): ELU(alpha=1.0)
    (2): Linear(in_features=512, out_features=512)
    (3): ELU(alpha=1.0)
    (4): Linear(in_features=512, out_features=Zs+64))
```

Hyperparameters used for DiffTOP for both model-based RL and imitation learning are shown in Tab 7. In model-based RL, we use the same parameters with TD-MPC (Hansen et al., 2022) whenever possible.

## C   ENVIRONMENT DETAILS

For model-based reinforcement learning evaluation, we use 15 visual continuous control tasks from Deepmind Control Suite (DMC). For imitation learning, we use 13 tasks (detailed information can

| Hyperparameter | Value |
|---|---|
| **Model-based RL** | |
| Max planning iterations | 100 (50) |
| Planning step size | 1e-4 (5e-3) |
| Discount factor | 0.99 |
| Action loss coefficient (c0) | 1 |
| optimizer | $Adam(\beta_1 = 0.9, \beta_2 = 0.999)$ |
| Gradient Norm | 10 |
| Planning horizon schedule | $1 \rightarrow 5$ (25k steps) |
| Batch size | 256 |
| Latent dimension | 50 |
| Sampling technique | $PER(\alpha = 0.6, \beta = 0.4)$ |
| Learning rate | 1e-3 |
| **Imitation Learning** | |
| Max planning iterations | 100 |
| Planning step size | 1e-4 |
| Planning horizon schedule | 1 |
| Latent dimension | 50 |
| Posterior Gaussian dimension | 64 |
| KL coefficien | 1 |
| Learning rate | 3e-4 |
| GMM Num Modes | 5 |
| RNN Seq Len | 16 |
| RNN Hidden Dim | 1000 |
| Point Cloud Sampled Points (ManiSkill) | 1200 |

Table 7: Hyperparameters used in DiffTOP.

be found in Table 8) from Robomimic (Mandlekar et al., 2021), IBC (Florence et al., 2022), ManiSkillp (Mu et al., 2021), and ManiSkill2 (Gu et al., 2023).

| Task | Source | Obs. Type | Ac Dim | Object | Demo | Task Description |
|---|---|---|---|---|---|---|
| Square | Robomimic | Img | 7 | Rigid | 200 | Pick a square nut and place it on a rod. |
| Transport | Robomimic | Img | 14 | Rigid | 200 | Transfer a hammer from a container to a bin |
| ToolHang | Robomimic | Img | 7 | Rigid | 200 | assemble a frame consisting of a base and hook |
| Push-T | IBC | Img | 2 | Rigid | 200 | Push a T-shaped object to a specified position |
| OpenCabinetDrawer | ManiSkill1 | Point Cloud | 13 | Rigid | 300/obj. | Open a specific drawer of the cabinet |
| OpenCabinetDoor | ManiSkill1 | Point Cloud | 13 | Rigid | 300/obj. | Open a specific door of the cabinet |
| PushChair | ManiSkill1 | Point Cloud | 22 | Rigid | 300/obj. | Push the swivel chair to the target position |
| MoveBucket | ManiSkill1 | Point Cloud | 22 | Rigid | 300/obj. | Move a bucket and lift it onto a platform |
| PickCube | ManiSkill2 | Point Cloud | 7 | Rigid | 1000 | Pick up a cube and move it to a goal position |
| Fill | ManiSkill2 | Point Cloud | 7 | Soft | 200 | Fill clay from a bucket into the target beaker |
| Hang | ManiSkill2 | Point Cloud | 7 | Soft | 200 | Hang a noodle on a target rod |
| Excavate | ManiSkill2 | Point Cloud | 7 | Soft | 200 | Lift a amount of clay to a target height |
| Pour | ManiSkill2 | Point Cloud | 7 | Soft | 200 | Pour liquid from a bottle into a beaker |

Table 8: Imitation Learning Tasks Summary.

We visualize the keyframes of the imitation learning tasks in Fig 8.

# D   MORE IMPLEMENTATION DETAILS ON USING CVAE FOR IMITATION LEARNING

We provide more details on how we instantiate DiffTOP with CVAE in imitation learning, in which the goal is to reconstruct the expert actions conditioned on the state. The CVAE encoder is composed of three networks: the first network is a state encoder $h_\theta^o$ that encodes the state into a latent feature vector $z^s = h_\theta^o(s_i)$, which is the conditional information in our case. The second is an action encoder $h_\theta^a$ that encodes the expert action into a latent feature vector $z^a = h_\theta^a(a_i^*)$. The last is a

fusing encoder $h_\theta^l(z^s, z^a)$ that takes as input the concatenation of the state and action latent features, and outputs the mean $\mu$ and variance $\sigma^2$ of the posterior Gaussian distribution $\mathcal{N}(\cdot|\mu, \sigma^2)$. During training, the final latent state $z$ for state $s_i$ used in Equation 7 is the concatenation of a sampled vector $\tilde{z}$ from the posterior Gaussian distribution $\mathcal{N}(\cdot|\mu, \sigma^2)$, and the latent state feature vector $z^s$: $z = [\tilde{z}, z^s], \tilde{z} \sim \mathcal{N}(\cdot|\mu, \sigma^2)$.

The latent state $z$ will then be used as input for the decoder, which consists of the reward function $R_\theta$, and the dynamics function $d_\theta$. Trajectory optimization is performed with the reward and dynamics function in the decoder to solve Equation 7 to generate the reconstructed action $a^*(\theta; s_i)$. The loss for training the CVAE is the evidence lower bound (ELBO) on the demonstration data:

$$\mathcal{L}_{DiffTOP}^{IL}(\theta) = \sum_{i=1}^{N} ||a(\theta; s_i) - a_i^*||_2^2 - \beta \cdot \text{KL}(\mathcal{N}(\cdot|\mu, \sigma^2)|\mathcal{N}(0, I)), \tag{9}$$

where $\text{KL}(P||Q)$ denotes the KL divergence between distributions $P$ and $Q$. At test time, only the decoder of the CVAE is used for generating the actions. Given a state $s$, the latent state $z$ is the concatenation of the encoded latent state feature $z^s$, and a sampled vector $\tilde{z}$ from the prior distribution $\mathcal{N}(0, 1)$.

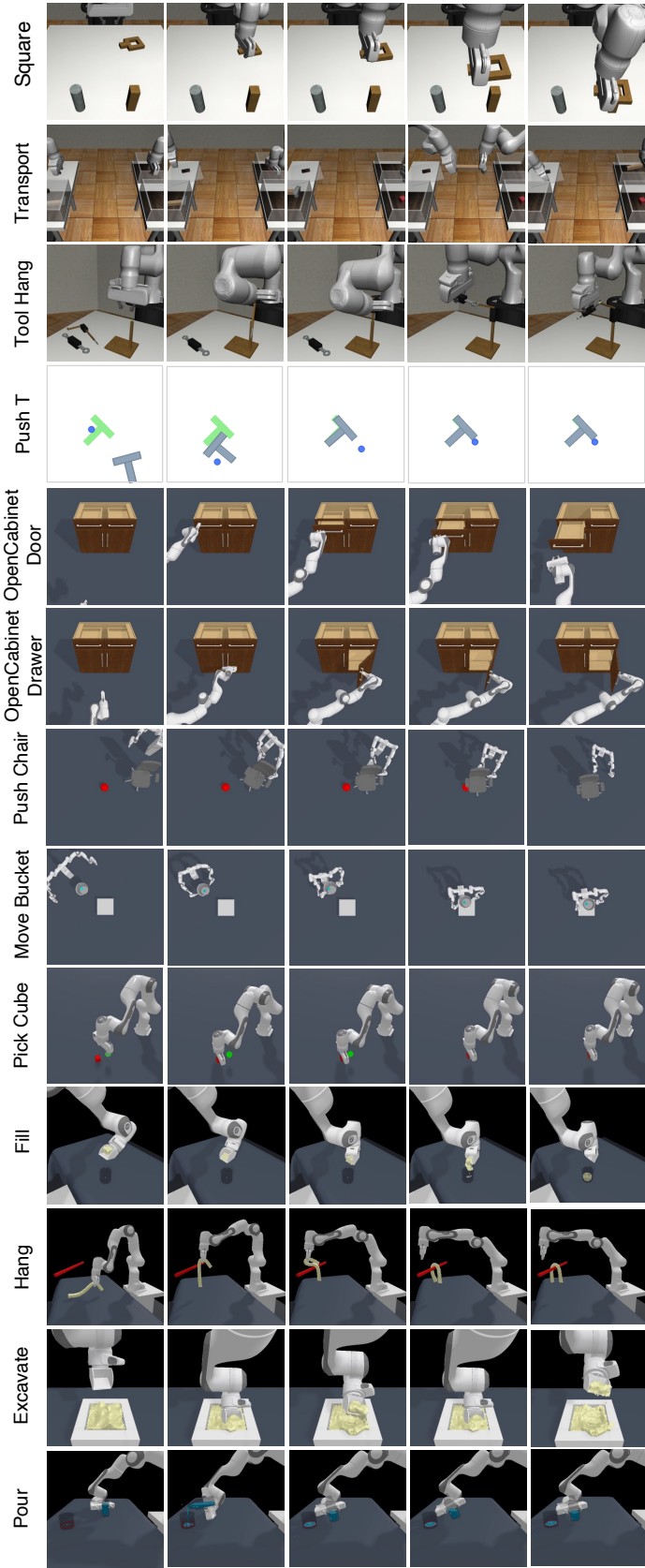

Figure 8: Visualization of the tasks for imitation learning.

