# OpenReview forum: "Differentiable Trajectory Optimization as a Policy Class for Reinforcement and Imitation Learning"
_ICLR.cc/2024/Conference — Submitted to ICLR 2024_

### Official Review · Reviewer_cY2u · 2023-10-22

**Soundness:** 4 excellent
**Presentation:** 4 excellent
**Contribution:** 4 excellent
**Rating:** 10
**Confidence:** 4

**Summary:**

The paper proposes DiffTOP, a novel policy class for sequential decision problems that is applicable to both reinforcement and imitation learning. The main idea is to use differential policy optimization to find efficiently a sequence of actions for a fixed starting state, given current estimates of the parameters that parameterize the observation-to-state encoder and the reward function and system dynamics in learned state space. The policy gradient is used to improve the estimates of the parameters, and because the computation of the action sequence is differentiable with respect to the parameters, the policy gradient ends up being differentiable with respect to the parameters, too.

**Strengths:**

The ability to compute policy gradients directly with respect to the parameters that describe the observation and transition model is a major advance, eliminating the need to do sample-based estimates. Another major strength, shared with the TD-MPC algorithm on which DiffTOP is based, is that model learning does not have to be done ahead of time, thus alleviating the model mismatch problem that is central to model-based reinforcement learning. A third strength of the paper is the very solid empirical verification of the approach on many control problems, both in the reinforcement and imitation learning setting.

**Weaknesses:**

It appears that the trajectory optimization solver the authors are using, Theseus, does not support constraint optimization, so they have to manually unroll the dynamics, instead of presenting it as a constraint to the optimizer, as is the usual practice in trajectory optimization. This does not look very convenient, but is probably only a limitation associated with the software currently being used, and might be resolved in the future by using a more advanced solver.

**Questions:**

Is the inability of Theseus to solve a constraint optimization version of the problem a fundamental limitation of how it works, or simply something not implemented yet? Maybe it solves non-linear least-squares problems analytically, and adding constraints would make this impossible?

---

> ### Author Response · Authors · 2023-11-21
> **Author Response**
>
> We thank the reviewer for their time and the valuable comments and suggestions. We address the reviewer’s questions as below.
>
> **Is the inability of Theseus to solve a constraint optimization version of the problem a fundamental limitation of how it works, or simply something not implemented yet? Maybe it solves non-linear least-squares problems analytically, and adding constraints would make this impossible?**
>
> We thank the reviewer for this question. As we are not the active developers of Theseus, we would not be able to answer this question with perfect accuracy. According to our understanding, Theseus solves the nonlinear least-squares problem numerically; specifically, we used the Levenberg–Marquardt algorithm in our implementation.
>
> The original Theseus paper has the following discussion about adding constraints in their limitation section: “The nonlinear solvers we currently support apply constraints in a soft manner (i.e., using weighted costs). Hard constraints can be handled with methods like augmented Lagrangian or sequential quadratic programs [99, 100], and differentiating through them are active research topics”. Based on this, it seems that adding constraints is not a fundamental limitation of how it works, but rather something that has not been implemented yet.

---

### Official Review · Reviewer_zYV2 · 2023-10-30

**Soundness:** 3 good
**Presentation:** 3 good
**Contribution:** 3 good
**Rating:** 8
**Confidence:** 4

**Summary:**

This paper leverages sensitivity analysis capabilities of trajectory optimization in order to differentiate the optimal solution with respect to problem parameters. This allows trajectory optimization to be used as a differentiable policy architecture, and the authors show that this has use cases in model-based reinforcement learning (MBRL) and imitation learning (IL). In MBRL, the latent-space encoder, latent-space dynamics, and the reward function parameters are learned within DiffTOP. In IL, the output of DiffTOP is forced to match the provided actions and DiffTOP carries out efficient inverse RL. The authors show that the proposed policy class outperforms baselines such as diffusion policies and IBCs.

**Strengths:**

1. Overall, the paper is well-written and the methods and results are easy to follow for people familiar with the existing methods.
2. The authors carry out extensive empirical testing across many baselines, the diversity of the considered benchmarks and baselines is quite appreciated.
3. The authors convincingly tell a story of objective mismatch in MBRL, and point out that DiffTOP is able to learn a representation that is not only better for minimizing dynamics error but also better for control. The contribution in the space of MBRL is quite strong.

**Weaknesses:**

1. How do the authors ensure that a reasonable answer is found in the process of nonlinear trajectory optimization with gradients? If the cost and dynamics was learned to formulate a convex trajectory optimization problem, this would not be a problem as the encoder is forced to learn a representation of that form. However the authors propose to use a fully nonlinear formulation and it's not clear how the training pipeline would be robust against finding bad local minima.

2. The contribution in the IL seems a bit weaker compared to MBRL. In the setting of imitation learning, it seems that the authors are performing a version of inverse RL (which similarly does not assume access to rewards, but learns rewards and dynamics from demonstrations). Making connections to previous methods within inverse RL would be good.

3. "For DiffTOP, we always use a base policy to generate the action initialization for the trajectory optimization problem" - this makes the comparison between other methods a bit unclear as other methods did not start with an informed initial guess. The experiment results make it seem like DiffTOP cannot be used as a standalone tool for IL, but rather a tool for the purposes of refinement; yet the method section is proposing a standalone IL algorithm.

**Questions:**

1. The introduction of CVAE to generate multimodal samples is reasonable, but there are many other routes that might potentially handle the multimodality problem. For instance, it seems cheaper and more efficient to consider a differentiable stochastic trajectory optimization framework where a small amount of noise can be added to the action at every step, which forces trajectory optimization itself to be a stochastic procedure that will generate multimodal actions. Have the authors considered this formulation?

2. CVAEs tend to be harder to train compared to recent diffusion-based generative models. Is it possible to train a diffusion model instead of address the multimodality problem? Maybe it is difficult since the author's framework requires latent states?

3. How does DiffTOP perform without informed initial guesses in the RoboMimic experiment?

---

> ### Author Response · Authors · 2023-11-21
> **Author Response**
>
> We thank the reviewer for their time and the valuable comments and suggestions. We address the reviewer’s questions and concerns as below.
>
> **How do the authors ensure that a reasonable answer is found in the process of nonlinear trajectory optimization with gradients? If the cost and dynamics was learned to formulate a convex trajectory optimization problem, this would not be a problem as the encoder is forced to learn a representation of that form. However the authors propose to use a fully nonlinear formulation and it's not clear how the training pipeline would be robust against finding bad local minima.**
>
> We thank the reviewer for this insightful question. Indeed, there is not much theoretical guarantee on the quality of the converged local minima when solving the nonlinear trajectory optimization problem. One thing we find particularly useful is to provide the nonlinear trajectory optimization problem with a good initialization. This is shown in our ablation study of model-based RL in Figure 4, where we compare the performance of DiffTOP, and DiffTOP without using the action initialization from the learned latent policy (in which case the action is initialized to be 0 in trajectory optimization) on 4 model-based RL tasks . The result shows that without the action initialization from the learned latent policy, the performance drops.
>
> The same results hold in the imitation learning setting, as shown in Table 3 & 4 in Appendix A.2  of the updated paper.  Following the reviewer’s questions, we have added experiments to test DiffTOP without informed initial guess in the RoboMimic and ManiSkill experiments in Table 3 & 4. As the results suggest, if the action is initialized to be zero or sampled from a Gaussian of N(0, 1), the performance of DiffTOP is similar or slightly worse to vanilla behavior cloning, possibly due to the convergence to bad local minimal during trajectory optimization. With better action initializations, i.e., using actions from BC-RNN or Diffusion Policy, the performance of DiffTOP improves.
>
> Therefore, a good practice of using DiffTOP is to provide it with a good action initialization. This is often feasible in practice: in model-based RL, as our experiments show, the learned latent policy can be used to provide this action initialization; for imitation learning, one could always first learn a base policy using any behavior cloning algorithm, and then use DiffTOP to further refine the actions from the base policy for better performances.
>
> We have updated Appendix A.2 of the paper to include the above discussion.
>
> **The contribution in the IL seems a bit weaker compared to MBRL. In the setting of imitation learning, it seems that the authors are performing a version of inverse RL (which similarly does not assume access to rewards, but learns rewards and dynamics from demonstrations). Making connections to previous methods within inverse RL would be good.**
>
> We thank the reviewer for this valuable comment. We agree that when applied to the IL setting, DiffTOP can be viewed as doing inverse RL as well. The differences between DiffTOP and previous IRL methods are as follows.
>
> In DiffTOP, the learned function is not necessarily a “reward” function as those used in a typical RL/MDP setting. It is just a learned “objective function”, such that optimizing it with trajectory optimization would yield actions that minimize the imitation learning loss with respect to the expert actions in the demonstration. We leave exploring the connections with inverse RL for future work.
>
> In contrast to our approach, inverse RL algorithms try to learn a reward in the MDP/RL setting, such that when optimizing the learned reward with RL, the policy’s state-action/state distribution matches the state-action/state distribution of the expert demonstrations. To perform such a distribution match, the loss for learning the reward is usually not the imitation learning loss as used in DiffTOP, but a loss to encourage matching the state-action/state distribution of the expert (e.g., the adversarial loss in GAIL [2]).
>
> We have updated the paper to include the connection to inverse RL at the end of section 5.2.1.
>
> [1]  Florence et al, Implicit Behavioral Cloning, CoRL 2021
> [2]  Ho et al, Generative Adversarial Imitation Learning, NeurIPS 2016

---

> > ### Author Response · Authors · 2023-11-21
> > **Author Response (continued)**
> >
> > **For DiffTOP, we always use a base policy to generate the action initialization for the trajectory optimization problem" - this makes the comparison between other methods a bit unclear as other methods did not start with an informed initial guess. The experiment results make it seem like DiffTOP cannot be used as a standalone tool for IL, but rather a tool for the purposes of refinement; yet the method section is proposing a standalone IL algorithm / How does DiffTOP perform without informed initial guesses in the RoboMimic experiment?**
> >
> >
> > We thank the reviewer for bringing this up. In our experiments, in addition to just comparing to other methods that do not start with an informed initial guess, we also compare to methods that use these informed initial guesses as well. They are “Residual+BC-RNN”, “IBC+Diffusion”, and “Residual+Diffusion'' in Table 1 (for RoboMimic), and “BC + Residual” in Table 2 (for ManiSkill). These methods use informed initial guesses as follow:
> >
> > For RoboMimic experiments:
> > Residual + BC-RNN: We use a pre-trained BC-RNN as the base policy, and learn a residual policy on top of it. The residual policy takes as input the action from the base policy, and outputs a delta action which is added to the base action. This is the most standard and simple way of doing residual learning.
> > IBC + Diffusion: A version of IBC that uses the action from a pre-trained Diffusion Policy as the action initialization in the test-time optimization process.
> > Residual + Diffusion: Similar to Residual + BC-RNN, but using a pre-trained Diffusion Policy as the base policy.
> >
> > For Maniskill experiments:
> > BC + residual: Learns a residual policy that takes as input the action from the BC policy and outputs a delta correction.
> >
> > As shown in Table 1, DiffTOP outperforms all these 3 methods on RoboMimic, and as shown in Table 2, DiffTOP also outperforms BC+residual on ManiSkill. This demonstrates that even with the same informed initial guess, DiffTOP is still the best method to refine the initial guessed actions.
> >
> > We have also added experiments on both RoboMimic and Maniskill to test the performance of DiffTOP without such informed initial guesses; instead, the actions are either initialized to be 0 or sampled from a Gaussian N(0, 1). The results are shown in Table 3 and 4 in Appendix A.2 of the updated paper. We do notice a drop in performance with random or zero-initialized actions, possibly due to the convergence to bad local minima during nonlinear trajectory optimization, as the reviewer mentioned before. We note that this would not be a drawback of applying DiffTOP in practice for imitation learning: one could always first learn a base policy using any behavior cloning algorithm, and then use DiffTOP to further refine the actions from the base policy for better performances. We have updated Appendix A.2 of the paper to include the above discussion.
> >
> >
> > **The introduction of CVAE to generate multimodal samples is reasonable, but there are many other routes that might potentially handle the multimodality problem. For instance, it seems cheaper and more efficient to consider a differentiable stochastic trajectory optimization framework where a small amount of noise can be added to the action at every step, which forces trajectory optimization itself to be a stochastic procedure that will generate multimodal actions. Have the authors considered this formulation?**
> >
> > We thank the reviewer for this suggestion! Stochastic trajectory optimization sounds very interesting. However, at this moment, we are not aware of any efficient and robust implementation of a differentiable version of stochastic trajectory optimization. We would definitely like to try it once there is a library for that, which we leave as interesting future work.
> >
> >
> > **CVAEs tend to be harder to train compared to recent diffusion-based generative models. Is it possible to train a diffusion model instead of address the multimodality problem? Maybe it is difficult since the author's framework requires latent states?**
> >
> > We thank the reviewer for this suggestion. We think it would definitely be possible to use a diffusion model instead of a CVAE to handle the multimodality problem, and given the recent success of diffusion in image generation, it might achieve even better results compared to CVAE. However, as the reviewer mentioned, it might require additional techniques/efforts to correctly combine diffusion with trajectory optimization in the latent space, which is beyond the current scope of the paper. We leave this as interesting future work.

---

> ### Comment · Reviewer_zYV2 · 2023-11-23
> **comment**
>
> I would like to thank the authors for the detailed response, I agree with the points in the rebuttal and will keep the score as is.

---

### Official Review · Reviewer_fm37 · 2023-11-01

**Soundness:** 3 good
**Presentation:** 1 poor
**Contribution:** 3 good
**Rating:** 6
**Confidence:** 3

**Summary:**

The paper proposes DiffTOP an algorithm that extends TD-MPC to include additional losses for imitation learning and reinforcement learning. The proposed algorithm is evaluated on many examples including the standard RL benchmark tasks as well as many imitation learning tasks.

**Strengths:**

The paper has a lot of evaluations on many standard benchmark tasks. The number of evaluations is very impressive in both the RL as well as the imitation learning domain. Furthermore, the algorithm seems to perform comparable or better to the TD-MPC baseline.

**Weaknesses:**

While the number of experiments is very impressive, the amount of content condensed into 9 pages is a bit too much. It seems like the authors wanted to press every little piece of information into the limited pages which drastically reduced readability. It would be great if the authors would go back to the drawing board and prioritize what is the important information and elaborate on these parts. For example, the algorithm is still very unclear to me, which model parameters exist and are optimized together. It would be great if the authors could introduce their algorithm more slowly and rely on less knowledge from the TD-MPC paper. While mentioning the relation to TD-MPC is important, requiring excellent knowledge of TD-MPC to read the paper is very limiting.

I am a bit torn on whether to score marginally above or below the acceptance threshold. While contribution and experimental evaluation are appropriate, the clarity of the presentation clearly lacks significantly and for me personally the clarity of the research is a major evaluation criterion.

**Questions:**

I have questions regarding the additional PG loss for the RL case. Does it make sense to optimize the model parameters w.r.t. the Q-function? Wouldn't this loss let the dynamics model prefer to hallucinate to obtain a high q-function? Learning the dynamics and reward model is a supervised learning problem while learning the q-function is not. Therefore, I would expect that the supervised learning problem is much easier and imagine that the additional information flowing through this additional PG loss is not that important. What is the intuition of the authors for the additional PG loss? For imitation learning, the motivation for the additional bc loss is much clearer.


**Post Rebuttal Comment:**

All good with me. I think the authors could still drastically improve the paper by reworking the writing. However, as the other reviewers are happy with the writing, it is fine with me. In the rebuttal the authors mainly added small band-aids to the text without rethinking the structure of the paper but I mean that is quite common for all rebuttals. I increased my score to weak-accept.

As there will inevitably be errors in the learned dynamics model, the additional PG loss regularizes the dynamics model to have low error in states and actions that lead to high rewards, i.e., those that are important for maximizing the task performance, instead of driving the dynamics model to hallucinate. Besides, we keep the dynamics prediction loss term when training the dynamics model, which should also prevent hallucination.

The sentence is quite good and should be added to the paper.

---

> ### Author Response · Authors · 2023-11-21
> **Author Response**
>
> We thank the reviewer for their time and the valuable comments and suggestions. We address the reviewer’s questions and concerns as below.
>
> **For example, the algorithm is still very unclear to me, which model parameters exist and are optimized together.**
>
> We have updated section 4.2 and 4.3 of the paper to make the model parameters more clear, and hope the following clarifications help with addressing this concern.
>
> At the beginning of Section 4.2, we have added the following sentence to more explicitly state the model parameters that are optimized: “We use $\theta$ to denote all learnable model parameters to be optimized in DiffTOP, including the parameters of the encoder $h_\theta$, the latent dynamics model $d_\theta$, the reward predictor $R_\theta$, and the Q value predictor $Q_\theta$” . Equation 6 in the paper illustrates the loss that is used to optimize these parameters.
>
> For imitation learning, we have added the following sentence at the end of section 4.3 to more explicitly show the model parameters that are optimized in this setting: “Similarly, We use $\theta$ to denote all learnable model parameters to be optimized in DiffTOP, which includes the parameters of the encoder $h_\theta$, the latent dynamics model $d_\theta$, and the cost function $f_\theta$ in the imitation learning setting”. Equation 8 in the paper shows the loss that is used to optimize these parameters.
>
>
> **It would be great if the authors could introduce their algorithm more slowly and rely on less knowledge from the TD-MPC paper. While mentioning the relation to TD-MPC is important, requiring excellent knowledge of TD-MPC to read the paper is very limiting.**
>
> We thank the reviewer for bringing this to our attention. Indeed, the model-based RL part of DiffTOP builds upon TD-MPC, thus requiring some knowledge of TD-MPC for understanding the paper. We do acknowledge that with the page limit, we are not able to cover every detail of the TD-MPC paper. However, we would like to clarify that we do have a section in our paper, i.e., section 3.2, model-based RL preliminaries, to explain the notations and external knowledge from TD-MPC that is needed for understanding our method, in order to make the paper self-contained. Following the reviewer’s suggestion, we have updated section 3.2 with more explanations of TD-MPC to add more buildup for introducing our algorithm, and for a better understanding of the paper.
>
> Please let us know if you have any more suggestions on how to improve the readability of the paper – we would be more than happy to further incorporate them.

---

> ### Author Response · Authors · 2023-11-21
> **Author Response (continued)**
>
> **I have questions regarding the additional PG loss for the RL case. Does it make sense to optimize the model parameters w.r.t. the Q-function? Wouldn't this loss let the dynamics model prefer to hallucinate to obtain a high q-function? Learning the dynamics and reward model is a supervised learning problem while learning the q-function is not. Therefore, I would expect that the supervised learning problem is much easier and imagine that the additional information flowing through this additional PG loss is not that important. What is the intuition of the authors for the additional PG loss? For imitation learning, the motivation for the additional bc loss is much clearer.**
>
> We thank the reviewer for this insightful question. Optimizing the model parameters with the additional PG loss through differentiable trajectory optimization is one of the core contributions of our paper (besides the imitation learning modification as the reviewer noted). The intuition behind this is as follows:
>
> As the reviewer mentioned, since learning the dynamics and reward model is a supervised learning problem that have been well-studied, existing model-based RL algorithms adopt this paradigm: they first learn a latent dynamics and reward model via supervised learning, and during inference time, they perform planning (such as MPPI) using the learned dynamics and reward model to generate the actions to take. However, recent studies [1, 2, 3] have shown that there is a fundamental “objective mismatch issue” with such a paradigm, i.e., models that achieve better training performance (e.g., lower MSE) in learning a dynamics/reward model are not necessarily better for control. These papers explain this phenomenon in more detail, but the intuition here is as follows: a learned dynamics model will typically not perfectly drive the dynamics loss to 0.  Two different learned dynamics models will have different types of errors, and the model with the lower MSE loss is not necessarily the model that will be most useful for optimizing actions to maximize reward; it depends on what states and actions are causing those errors and what types of errors they are.  For example, errors in predicting the dynamics for actions that lead to low rewards are not very important.
>
> DiffTOP addresses this issue: by computing the PG loss on the optimized actions from trajectory optimization and differentiating through the trajectory optimization process, the dynamics and reward functions are both optimized directly to maximize the task performance.
> As there will inevitably be errors in the learned dynamics model, the additional PG loss regularizes the dynamics model to have low error in states and actions that lead to high rewards, i.e., those that are important for maximizing the task performance, instead of driving the dynamics model to hallucinate. Besides, we keep the dynamics prediction loss term when training the dynamics model, which should also prevent hallucination. In our experiments, DiffTOP with the added PG loss greatly outperformed the TD-MPC baseline that does not have this loss term, demonstrating its effectiveness.
>
>
> [1] Lamber et al, Objective mismatch in model-based reinforcement learning, Learning for Dynamics and Control (L4DC), 2020
> [2] Eysenbach et al, Mismatched no more: Joint model-policy optimization for model-based rl, NeurIPS 2022
> [3] Ghugare et al, Simplifying model-based rl: learning representations, latent-space models, and policies with one objective, ICLR 2022

---

> > ### Author Response · Authors · 2023-11-22
> > **Official comment by authors**
> >
> > We thank the reviewer again for their time and effort helping us improve our paper!
> > Since the rebuttal period is approaching its end, we would really appreciate it if the reviewer could let us know if their concerns have been addressed, and if we can provide additional clarifications to improve our score.

---

### Official Review · Reviewer_WGnQ · 2023-11-10

**Soundness:** 4 excellent
**Presentation:** 4 excellent
**Contribution:** 3 good
**Rating:** 8
**Confidence:** 4

**Summary:**

The paper introduces DiffTOP, a novel policy class for reinforcement learning (RL) and imitation learning (IL) that employs differentiable trajectory optimization to generate policy actions. This approach leverages recent advancements in differentiable trajectory optimization, allowing end-to-end learning of cost and dynamics functions through gradient computation. DiffTOP addresses the "objective mismatch" problem in model-based RL by optimizing the dynamics and reward models to directly maximize task performance. For imitation learning, it outperforms previous methods by optimizing actions with a learned cost function at test time.

The authors benchmark DiffTOP on 15 model-based RL tasks and 13 imitation learning tasks with high-dimensional inputs like images and point clouds. The results show that DiffTOP surpasses prior state-of-the-art methods in both domains. The paper also includes an analysis and ablation studies to provide insights into DiffTOP's learning procedure and performance gains.

In summary, the contributions of the paper are:

1) Proposing DiffTOP, which uses differentiable trajectory optimization for RL and IL.
2) Demonstrating through extensive experiments that DiffTOP achieves state-of-the-art results in both RL and IL with high-dimensional sensory observations.
3) Providing analysis and ablation studies to understand the learning process and performance improvements of DiffTOP.

**Strengths:**

The paper presents a robust and technically rigorous study, bolstered by a comprehensive suite of experiments and ablation studies. Some of the experiments are performed in notably challenging environments, showcasing the strength of the proposed method. It addresses the complex "objective mismatch" problem inherent in model-based reinforcement learning. The overall style of the paper is commendable, with clear and detailed descriptions of the experimental setup, network architectures, and hyperparameters, as well as well-structured pseudocode.

**Weaknesses:**

No big weaknesses overall. It would be good to see the comparison of the computational efficiency of given algorithms and plots for return vs wall-clock time in addition to the number of samples.

**Questions:**

1) Are there any cases when TD-MPC performs or could theoretically perform better than DiffTOP?
2) What’s the computational cost of using Theseus for differentiable trajectory optimization vs MPPI?
3) Could you share the training plots of returns vs wall-clock time at least for some of the environments? It would be useful to compare not only the sample but also the computational efficiency of the different algorithms.

---

> ### Author Response · Authors · 2023-11-21
> **Author Response**
>
> We thank the reviewer for their time and the valuable comments and suggestions. We address the reviewer’s questions as below.
>
> **It would be good to see the comparison of the computational efficiency of given algorithms and plots for return vs wall-clock time in addition to the number of samples / Could you share the training plots of returns vs wall-clock time at least for some of the environments? It would be useful to compare not only the sample but also the computational efficiency of the different algorithms.**
>
> We thank the reviewer for this suggestion. We have added plots for return vs wall-clock time of TD-MPC and DiffTOP on some of the model-based RL tasks, shown in Figure 7 in the updated paper, along with a discussion of the results in Appendix A.1.
>
> To provide a short summary here for the reviewer’s convenience:
> In terms of computational efficiency (e.g. wall-clock time needed to achieve a given level of performance), the results are environment-dependent. Compared to TD-MPC, DiffTOP is more computationally efficient on 3 environments, has similar computational efficiency on 3 environments, and worse computational efficiency on 3 environments (please see Figure 7 of the updated paper). The main advantage of our method is not computational efficiency but sample efficiency, as shown by our superior performance in Figure 3.
>
> Compared to TD-MPC, it takes more wall-clock time for DiffTOP to finish the training of 1M environment steps. The major reason is that solving and back-propagating through the trajectory optimization problem is slower than using MPPI as in TD-MPC. As a reference, it takes 0.052 seconds to use Theseus to solve and differentiate through the trajectory optimization problem, and 0.0092 seconds to use MPPI as in TD-MPC.
>
> The research community is actively developing better and faster algorithms/software libraries for differentiable trajectory optimization, which could improve the computation efficiency of DiffTOP. For example, in all our experiments, we used the default CPU-based solver in Theseus. Theseus also provides a more advanced solver named BaSpaCho, which is a batched sparse Cholesky solver with GPU support. When we switch from the default CPU-based solver to BaSpaCho, the time cost of solving the trajectory optimization problem is reduced by 22% from 0.052 seconds to 0.041 seconds.
>
>
>
> **Are there any cases when TD-MPC performs or could theoretically perform better than DiffTOP?**
>
> We thank the reviewer for this interesting question. Empirically, as shown in Figure 3 in the paper, DiffTOP always performs better than or on par with TD-MPC. Theoretically, if the latent dynamics model and the latent reward model is perfectly learned without any error, i.e., if they exactly match the ground-truth dynamics and reward model, and if enough samples are used in MPPI,  TD-MPC should be able to achieve the maximal possible performance and DiffTOP would not be able to further outperform it. However, in practice, since there will always be errors in the learned latent dynamics and reward model, DiffTOP outperforms TD-MPC by directly learning a latent space that is useful for control, instead of just minimizing the dynamics or reward prediction error.
>
> **What’s the computational cost of using Theseus for differentiable trajectory optimization vs MPPI?**
>
> We thank the reviewer for this question. We have performed an analysis on the computation cost, and updated the paper to have a discussion about this in Appendix A.1. The results are summarized as follows:
>
> For inferring the action at one time step, the computation cost of using the default CPU-based solver in Theseus for differentiable trajectory optimization is 0.052 seconds. For MPPI, the time cost is 0.0092 seconds. However, as mentioned before, the computation cost of DiffTOP can be further reduced with better algorithms/solvers. For example, if we switch from the default CPU-based solver in Theseus to a more advanced batched sparse Cholesky solver with GPU support, the computation cost of solving the trajectory optimization problem is reduced by 22% from 0.052 second to 0.041 second. As the community is actively developing better libraries/algorithms for differentiable trajectory optimization, we believe the computational efficiency of DiffTOP would further improve in the future.

---

### Public Comment · ~Arun_Kumar_Singh1 · 2023-11-14
**Attention to a closely related work**

Dear Authors

I found this paper very interesting and insightful. We have also been working on similar problems. In this context, I would like to draw your attention to our recent work that was published in IROS 2023. The arxiv version of our work can be found in (https://arxiv.org/pdf/2310.14766.pdf). I see a lot of similarity in ideas between our work and yours, although we applied our work only in the context of navigation. Nonetheless, please consider including our work in your "Related Works" Section as I believe it is the closest existing work to what has been proposed in your paper.

The following are some of the similarities between our work and your proposed work.

1. Embedding a differentiable optimizer with CVAE: We embeded a custom constrained differentiable optimizer within CVAE encoder-decoder architecture to simultaneously learn some set-points in the cost-terms as well as initialization to accelerate the downstream constrained optimizer (Section III-B, Remark 1).

2. We also leverage the multi-modality resulting from the embeded optimizer within the decoder of the CVAE to generate diverse trajectories around obstacles. Essentially, our trajectory outputs looked very similar to Fig.5 of your paper.

---

> ### Author Response · Authors · 2023-11-21
> **Response**
>
> Thank you for your interest in our paper, and thanks for drawing our attention to your paper.
> As your paper is released on arxiv on Oct 23, 2023, which is past the ICLR submission date, we are not aware of it at the time of submission. It looks very relevant indeed, we will read it in more detail, and include it in the related work in the updated version.

---

> > ### Public Comment · ~Christopher_Diehl1 · 2023-11-29
> > **Further related work**
> >
> > Dear Authors,
> >
> > I also found the idea of this paper and the extensive evaluation across multiple environments very interesting. Following up on the previous discussion, I would also like to kindly draw your attention to closely related works [1, 2, 3] that use differentiable optimization as a policy class for imitation learning by utilizing Theseus.
> >
> > The works all differentiate through an optimization problem, with learned costs/energies and action initializations. While [1] performs a single-agent optimization, [2] and its extended version [3] parametrizes the cost with a game-theoretic multi-agent formulation. The similarity to your work can also be observed when comparing Fig. 2 of your work with Fig. 1 of [2,3].
> >
> > In contrast to your work, the mentioned papers do not learn the latent dynamics, which is an interesting extension. Moreover, [1-3] are only evaluated in mobile robots/ autonomous driving environments with an object-based input representation. In contrast, your work covers a broader spectrum of experimental environments and additionally uses image and point cloud inputs. Lastly, [2] and [3] also account for multimodality by parallel optimizations with different learned initializations. In your work, this is achieved by using a CVAE, whereas [2] and [3] use a min-of-k loss.
> >
> > It should be noted that [1] was published on arXiv in 2022, whereas [2] was published on OpenReview closely before the ICLR submission deadline.
> >
> > Thank you for your consideration. Maybe some of the ideas in the mentioned papers will also be helpful for future work.
> >
> > Kind Regards
> >
> > [1] Zhiyu Huang, Haochen Liu, Jingda Wu, Chen Lv, "Differentiable Integrated Motion Prediction and Planning with Learnable Cost Function for Autonomous Driving", IEEE Transactions on Neural Networks and Learning Systems", 2023
> >
> > [2] Christopher Diehl, Tobias Klosek, Martin Krueger, Nils Murzyn, Torsten Bertram, "On a Connection between Differential Games, Optimal Control, and Energy-based Models for Multi-Agent Interactions", International Conference on Machine Learning, New Frontiers in Learning, Control, and Dynamical Systems Workshop, 2023
> >
> > [3] Christopher Diehl, Tobias Klosek, Martin Krueger, Nils Murzyn, Timo Osterburg, Torsten Bertram, "Energy-based Potential Games for Joint Motion Forecasting and Control", Conference on Robot Learning 2023

---

> ### Public Comment · ~Arun_Kumar_Singh1 · 2023-11-21
> **Thanks for the reply**
>
> The paper was accepted in July. We were slightly late in pushing to Arxiv. But we would really appreciate it if you can include it because it is indeed the closest related work to yours

---

### Author Response · Authors · 2023-11-21
**Summary of change to the paper for rebuttal**

We thank the reviewers for their time and the valuable comments and suggestions. We have made several updates to the paper to address the reviewers’ comments. All changes are marked as blue in the updated paper. The updates are summarized as follows:

- We have updated section 3.2 with more explanations of TD-MPC to add more buildup for introducing DiffTOP, and for a better understanding of the paper (reviewer fm37).
- We have updated section 4.2 and 4.3 to make the model parameters that are being optimized in DiffTOP to be more clear (reviewer fm37).
- We have updated section 5.2.1 to include a discussion of the connection between the imitation learning version of DiffTOP and previous IRL methods (reviewer zYV2).
- We have added Figure 7 in Appendix A.1 to show plots for return vs wall-clock time, along with a discussion on the computational efficiency of DiffTOP, and computation costs of Theseus in Appendix A.1 (reviewer WGnQ).
- We have added new experiments to show the performance of DiffTOP without informed initial guesses for the imitation learning experiments on RoboMimic and Maniskill, summarized in table 3 and table 4 in Appendix A.2, along with a discussion on the results (reviewer zYV2).

---

### Meta-Review · Area_Chair_e48K · 2023-12-12

**Metareview:**

This paper proposes to differentiate through the solution of an optimal control problem, with applications to policy gradient learning and imitation learning. All reviewers were very positive about the paper.

Unfortunately, the paper completely disregards the existing literature.

Using Levenberg-Marquardt on an optimal control problem is called Iterative Linear Quadratic Regulator (ILQR). It is possible to differentiate through it using autodiff software, see e.g., https://github.com/google/trajax#trajectory-optimization-and-optimal-control. There is also iLQR, which resembles Levenberg-Marquardt, but works better.

Using differentiable trajectory optimization within policies is certainly not new (see references below) nor is the idea of pairing the differentiable trajectory optimization problem with learnable value functions. A second issue is the limited testing. The authors are only doing the standard MUJOCO simulation tasks, and have either infinitesimal improvement or worse performance.

Existing papers:
- https://arxiv.org/abs/1810.13400: "Differentiable MPC for End-to-end Planning and Control", Amos et al, 2018
- https://arxiv.org/abs/2310.14766: "End-to-End Learning of Behavioural Inputs for Autonomous Driving in Dense Traffic", Shrestha et al, 2023
- https://arxiv.org/abs/2209.10780: "Learning Model Predictive Controllers with Real-Time Attention for Real-World Navigation", Xiao et al, 2022
- https://arxiv.org/abs/2310.14468: "Revisiting Implicit Differentiation for Learning Problems in Optimal Control", Xu et al, 2023 (see their related work section for NUMEROUS existing references)

Because the paper completely ignores existing works, I do not recommend acceptance in the current state. The paper currently reads as if the idea is new, which is very misleading.

**Justification For Why Not Higher Score:**

Differentiating through an optimal control problem is well-known. The paper completely ignores the literature.

I think the reviewers who gave a really high score were not familiar with the field. They probably got misled by the paper's tone, which makes it sound like the idea is ground breaking

**Justification For Why Not Lower Score:**

N/A

---

### Decision · Program_Chairs · 2024-01-16

Reject